# Shaping Landscapes: Thinking On the Interactions between People and Nature in Inter- and Postdisciplinary Narratives

Ana Cristina Roque [1], Cecilia Veracini [2,*] and Cristina Brito [3]

[1] CH-ULisboa—Centro de História, Faculdade de Letras, Universidade de Lisboa, 1600-214 Lisboa, Portugal; anaroque1@campus.ul.pt

[2] CAPP—Centre for Public Administration & Public Policies, School of Social and Political Sciences, Universidade de Lisboa, Rua Almerindo Lessa, 1300-663 Lisboa, Portugal

[3] CHAM—Centre for the Humanities, Faculdade de Ciências Sociais e Humanas, FCSH, Universidade NOVA de Lisboa, 1069-061 Lisboa, Portugal; cbrito@fcsh.unl.pt

[*] Correspondence: cveracini@iscsp.ulisboa.pt

**Abstract:** This article addresses broad and plural concepts of landscape, considering its diversity of meanings and uses, which go far beyond its environmental and geographical connotations. It discusses the relationship between humanity and the rest of the natural world as a global process that combines physical and cultural aspects, and it seeks to highlight the contribution of environmental humanities to the understanding of these. Given the multiple conceptual interpretations and meanings of landscapes, we argue that current research trends are good examples of what we can consider as postdisciplinary approaches, challenging both disciplinary and interdisciplinary models of analysis. In this context, we use the recent pandemic scenarios as an example.

**Keywords:** environmental humanities; environmental history; landscapes and disciplines; human–nature relationships





## 1. Landscapes and Environmental Humanities: Relationships and Connections

Over the centuries, different aspects of the relationship between the human and the natural world have shaped a wide range of environments and landscapes. In the broad sense, landscapes—both physical and cultural ones—mirror the synthesis of interactions between peoples and places, reflect construction and circulation of knowledge and technology and materialize the development, transformation, and adaptation of human societies across time and space, in different geographical and cultural contexts. The result of these complex and multifaceted interconnections is the recognition of different spaces and settings as a structural component of the natural, historical, cultural, and scientific heritage and a vital element in the creation of each community's identity. Therefore, in a broader perspective, we assume that the concept of landscape is more than a unit of geography used to characterize the geographical association of facts (Sauer 1925). As a plural concept, it also encompasses worldviews and cultural views (e.g., Sepie 2017; Henry 2018), mindsets, also called mindscapes (Jacobs 2006), literary landscapes (Sutherland 2018) and even soundscapes (Rudi 2011; Miller 2013).

A new and wider vision of the concept of landscape would make it possible to go beyond the traditional scheme of disciplinary separation and would open a more fruitful discussion on current topics through the integration of the various components and actors involved. The concept of landscape can thus become a means to address the most urgent environmental issues that societies are facing today. In turn the emergence of the environmental humanities, which connect different fields of human and social sciences in order to address environmental issues and paradigms, is part of the growing willingness to engage with the environment from a new perspective. It represents an effort to enrich environmental research with a broader conceptual vocabulary and to rethink the

exceptionalism of the human being (e.g., Rose et al. 2012; Iovino et al. 2018; Thornber 2016; Little 2017; Holm and Brennan 2018; Roque et al. 2020). The environmental humanities, joined with a new landscape approach, will make it possible to go beyond regional and national portraits and consider new transcontinental and global scenarios far beyond westernized categories, at the same time revaluing traditional and indigenous worldviews (Sepie 2017; Emmanouilidou and Toska 2020). Likewise, this can allow the deconstruction of physical geographies in order to consider the world globally without arbitrary borders and, consequently, to reflect the dynamics of the different species that inhabit it. As argued by Berque (2000, p. 4), "landscape is not the environment, but has a certain relation with it. The question is thus to determine whether such a relation exists or not".

Additionally, humanistic environmental studies focus largely on different cultural products ranging from architecture, literature, poetry and nonfiction writing, drama, music, visual arts, films, and other media, to eco-criticism, politics, history, religion, philosophy, medicine, social and natural sciences (Thornber 2016; Roque et al. 2020). These studies stem from the work of scholars with different backgrounds and stimulate cultural responses to answer today's challenges. Therefore, bringing together scholars and methodologies from different scientific areas makes us aware of the diverse perspectives on human/non-human relationships (Thornber 2016) and of how these interactions have shaped and changed landscapes according to different perceptions and realities (Pasca et al. 2019).

This article is intended as a stimulator of thought about landscape concepts. We aim to underline the importance of an inter- and postdisciplinary approach to the concept of landscape in western history and culture, which can act as a starting point for future discussions about relationships between humans and nature. This work does not pretend to be exhaustive (and) or to include all the meanings of the concept of landscape, nor can it cover the different designations that this concept takes on in different regions and cultures of the world. It is a starting point for this discussion, designed to provide insights and to provoke thinking, responses and creative solutions to some of our environmental and societal issues.

## 2. Interdisciplinary and Postdisciplinary Approaches to Landscapes

Landscapes can be perceived as spaces for relationships and connections (Braun and Knitter 2016), as spaces of interface between different realities such as land and water, and in this case they can be referred to as seascapes,[1] waterscapes, or aquascapes (Richter 2015; Brito et al. 2019). Interfaces and frontier areas, usually modeled by conceptual binaries—e.g., land/sea, water/air, close/distant, human/animal, culture/nature, beautiful/unsightly—eliminate disciplinary barriers in order to find new narratives, about environmental history (e.g., this issue, Veracini 2020; Carvalho 2020) or ecocriticism and literary studies (this issue, Spini 2020). However, they can also overcome these limits and intermingle views and perspectives (e.g., this issue, Wickberg and Gärdebo 2020). Other authors, working on the semiotic concept of landscape, developed the idea of an 'ecosemiotic' associated to landscape (Farina 2010; Lindstrom et al. 2011). In particular, Farina (2021) considers a landscape "as an ecosemiotic system of geographic, ecological, cultural, contemplative and aesthetic signs and symbols" and this interpretation "can create a condition for a new union between the humans and the natural world". The seascapes of Cape Verde Islands captured by Noronha's camera are an example (see Figure 1).

---

[1] Hiroshi Sugimoto's Seascapes: https://www.sugimotohiroshi.com/seascapes-1 (accessed on 1 May 2021).

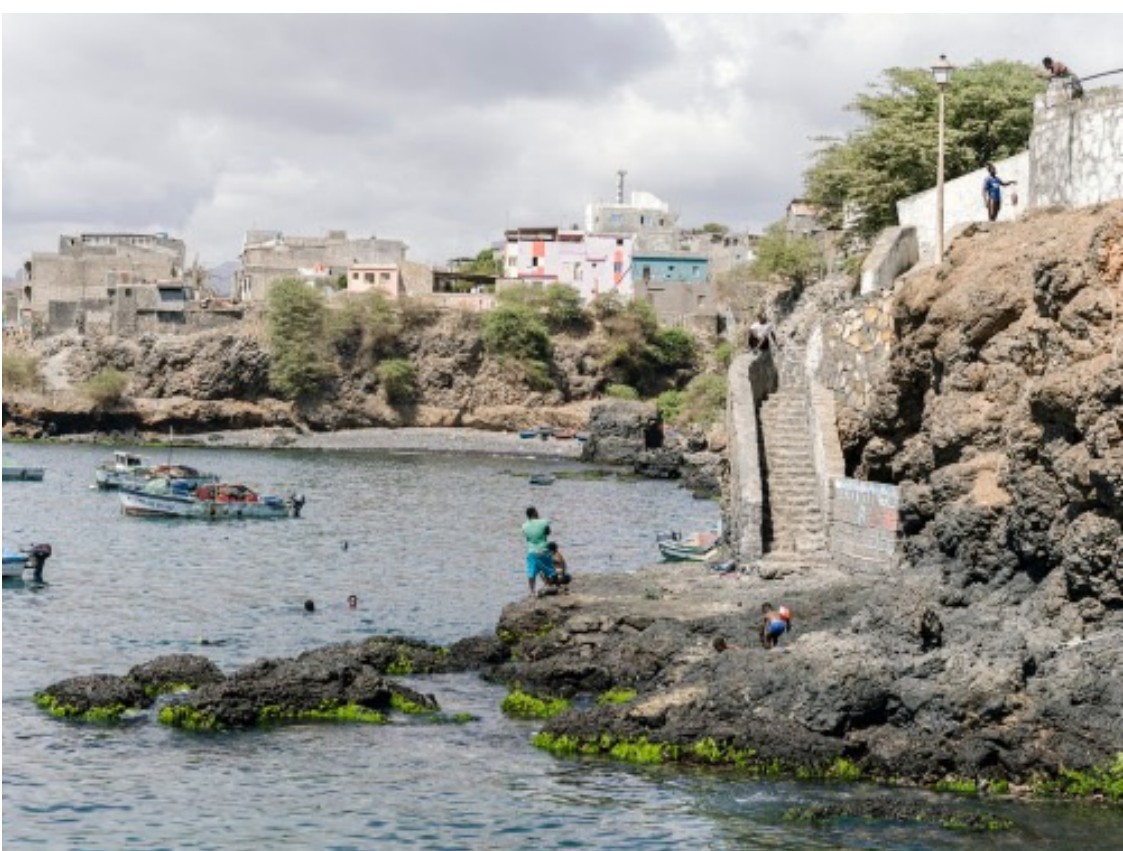

**Figure 1.** A seascape of the Cape Verde Islands, photographed by Hermano Noronha, from the exhibition '*Pai Mar*' (Father Sea) on magazine.landscapes.net. See also Vieira et al. (2020).[2]

Through depicting Cape Verde seascapes, formed both by people and the ocean, Noronha clearly shows that this landscape is not limited to what is visible (Berque 2000 and highlights how disciplinary boundaries can be deconstructed. The seascapes of this African archipelago can also be seen as places where not only people but the history, heritage and physical presence of whales, and moreover, intangible memories, engage to co-construct an historical and current identity (this issue, Vieira et al. 2020). Images can play an important role in changing individual behavior and public policy; there is a potential powerful influence in art production and viewing that can boosts awareness and stimulate actions. Similarly, literature (poetry, short stories, novels, as well as memories and oral traditions) can help to understand humans' relationships with nature and space and to create new forms of interactions (Merchant 2020).

The history of whaling and the people–whale relationship, for instance, can be presented to the reader from the animal's point of view, placing them as agents, actors and narrators of a story and a landscape built in close interaction (O'Brian 2017; Brito 2019). From a past of exploitation of whales and dealing with the strandings of these large cetaceans, to keeping these alive as a memory in the form of art and museum pieces and exhibitions, to capturing their essence in photography, it is possible to address a living and changing littoral that is mediated through marine animals and peoples' perceptions about it. They emerge not just in history, but also in marine science and environmental conservation, in literature and poetry, in teaching and dissemination. They show us that a past of the appropriation of nature has been left behind, and a change has occurred, transforming the seashore into a place of convergence (Vieira et al. 2020).

---

2   https://landscape-stories.tumblr.com/tagged/Pai-Mar (accessed on 1 May 2021). This is an output of the European project MSCA-RISE-777998 CONCHA—The construction of early modern global cities and oceanic networks in the Atlantic: An approach via the ocean's cultural heritage.

Seascapes and waterscapes, but also the colonial and postcolonial landscapes imbedded in this landscape, have become, as such, arenas where the traditional concepts of understanding and utilization might be questioned. Aspects of geography, settlement, urbanism, and architecture can be addressed, as well as people's feelings, their ways of living, their impact on changing environments and transformation of ecosystems, on natural populations of fauna and flora, and multiple interspecific interactions. These examples show a variety of opportunities for the construction of other narratives where the landscapes no longer have purely geographical connotations.

Another example is that of soundscapes, still an elusive and provocative concept, being continually (re)defined by different scholars. Despite the fact these rely on the assumption that both human history and the history of nature are full of sonic elements and interactions, they all have a purpose and meaning (Kelman 2010). The word soundscape refers to the total amount of sounds that can be heard at any moment in any given place (Rudi 2011) and which can only be understood through sensual perceptions (Miller 2013); studies on soundscape have become particularly relevant, as they concern both natural and artificial sound environments and attempt to understand their importance, and why sounds matter.

Clearly the word landscape takes on several meanings, but the most common is to evoke a feeling of awe-inspiring, creative, and almost poetic contemplation, as is usually depicted in paintings, photography, films, literature or poetry. It can appeal to a sense of belonging and admiration, of well-being and health (e.g., Tyson 1998), but also of fear and dismay (e.g., Freitas 2016) or even work (Morse 2010). As Cosgrove (1985) noted, the holistic and subjective implications of the idea of landscape are at the origin of the appropriation of a geographical concept that has now been applied to different realities. These realities depend on several factors, such as sensations, emotions, climatic conditions, a specific moment in a determined time and place, or even the physical environment to which we refer, and the investment made by each one in its 'construction' related to individual or collective expectations. In such a context, perceiving the different landscapes is always performing an act of remembrance (Ingold 2002).

Popular culture gives landscapes important meanings and interpretations. It is a way for people to access 'structures of feeling' that characterize a society at a particular sociocultural and historical juncture (Bulfin 2017; e.g., this issue, Kim 2020) and thus also for them to play a central role in building a common sense of place and quality of life (Morse 2010; this issue, Nuttall 2020). In addition, not only is there a wide variety of possible forms of landscapes, but the formats for publishing these studies to peers and reaching out to the public at large can be innovative or even disruptive. We can already find a myriad of formats, ranging from the traditional books and papers published in journals to online exhibitions (ENHANCE ITN 2019), art installations and poetic readings ( Woodcock 2015).

The chronologies of landscapes are equally vast, as vast as is the influence of individuals and peoples in their home places. We can refer to local, traditional, or indigenous landscapes, as well as to colonized landscapes (e.g., Hollsten 2008) and postcolonial ones (e.g., Carney 2017). On the other hand, if we consider Jacobs' theory of the tripartite landscape related to reality(ies) (Jacobs 2006), landscapes appear as different phenomena depending on the physical, social or inner reality they represent and reflect, which allows us to consider the existence of matterscapes (the landscape in its physical or environmental reality), powerscapes (the landscape in its social reality or culture) and mindscapes (the landscape in its inner reality, which is constituted by consciousness or states of mind). Thus landscapes, tangible and intangible, are all places or settings which humans act(ed) upon and, in some way, transform(ed).

Within this broad perspective, landscapes can also be discussed in terms of knowledge, and how knowledge is apprehended and validated according to both disciplinary rules and interdisciplinary views. As a complex and somehow ambiguous concept, landscapes can be perceived, seen, described, and represented in a multitude of layouts and sources, and be studied by researchers adhering to (apparently) opposite scientific and disciplinary

poles, from cultural studies, environmental history and ecocriticism (e.g., Bergman 2012) to ecological, anthropological and climate change investigation. However, what seems most interesting and innovative in landscape studies is the fact that they can be developed starting from an initial approach that combines different disciplines (e.g., Henry 2018; Brito 2019) and methodologies.

This particular characteristic makes landscape studies an example of interdisciplinary studies and a suitable subject for testing postdisciplinary approaches where barriers can be broken and bridges between traditional divisions can be built. In the past 20 years "the sciences have been moving toward organizing their practitioners around *problems*, not disciplines, in clusters that may be too short-lived to be institutionalized into departments or programs or to be given lasting disciplinary labels" (Biagioli 2009, p. 819). Postdisciplinary science derives from the realization that some urgent problems or issues cannot be adequately addressed in a single discipline. This type of approach should encourage people to question and seek to solve current problems, as well as to reconsider ways of addressing old and poorly solved problems, especially those that are more sensitive to conventional approaches (*Nature* 2015). Reorganizing institutions and people around specific issues or problems and abandoning disciplinary boundaries could become a more effective method of dealing with challenging issues.

Recent postdisciplinary approaches to knowledge advocate the concept of 'transscape', which focuses on how the different perception and cognition of a viewer brings about changes in the appearance of a landscape, and how these changes can be incorporated into the design of the world we see. "Landscape does not always refer to visual shapes, but involves symbols, in which something invisible turns into something visible and vice versa, repeatedly going through a cycle of appearance and disappearance, while transforming itself little by little" (Hanamura 2019, p. 44). The landscape itself indicates neither geographical features, nor trees nor constructions, yet it contains all of these. In this sense the term 'landscape' is different from 'scenery'. The latter word gives an objective description, whereas the former word, landscape, cannot be separated from the perspective of a viewer, or a subjective setting. Thus, it is the relationship between the land and the viewer's eyes (scape) that creates a landscape.

'Sea Change/Sea Edge', developed within a Swedish international research group, is a postdisciplinary knowledge and capacity-building project with the overall goal of deepening ecological understanding and culturally contextualizing scientific insight in ecofeminist theory, posthumanities and coastal communities. This partly submerged environmental arts and humanities project turns our attention and appreciation to low-trophic creatures—seaweeds, kelp forests, oysters, mussels and other -, to the habitat of the tidal zone and to mariculture (Asberg 2020). It stimulates new societal responses, opens up room to cultural imagination and invites a sea change in terms of peoples' perception and actions on the state of sea life. It aims to reconstruct the littoral seascapes, in a postdisciplinary approach that connects marine sciences, natural history, cultural heritage and sustainability engineering with research into both the arts and feminist environmental humanities (Asberg 2020).

The edge of the sea remains a strange and beautiful place, with all its wondrous creatures (from algae to whales), providing a host of benefits for various organisms, humans included. In other cultures, waterscapes can be viewed and conceptualized by the existing relationships of humans with the rest of nature. More than places of connection, they can be framed as places of partnerships, as kin or as members of a wider family that includes and affects both people and elements of nature at large (e.g., Krenak 2019, in a piece the title of which can be translated as 'The unsustainable embrace of progress or once upon time there was a forest in *Rio Doce*').

These entangled ecologies, cultures, realities and memories (Krenak 2019; Asberg 2020; Vieira et al. 2020) allow humans to reconnect with these landscapes and to reimagine futures and transformative practices of environmental humanities. Within future landscapes, art, science and society intermingle.

### 3. Future Landscapes

Our world is changing continuously and in 2020 it changed dramatically. The great impact of the new coronavirus outbreak has changed nearly every aspect of our daily life, significantly altering landscapes around the world. Grounded aircraft, deserted beaches, quieter streets, and new human behavior marked by social distance became normal sights during the early and final months of 2020. This new pandemic landscape and the causes that have produced it raise new questions on a global level and demand new answers, so this is a good moment to address new and multiple concepts of landscape. In the months of lockdown we have had time for a moment of introspection into our personal and intimate world, which perhaps we may have lost in some cases; and this 'personal archaeology' could act as a new tool through which we can approach global changes. The violence of this event has far surpassed previous situations and brought to national and international agendas the importance of accelerating the recovery of the lost balance between human beings and nature, highlighting the absolute need for a new compromise with the natural world (Ellwanger et al. 2020; Greger 2007; Asberg 2020). Furthermore, it has shown that philosophers, scholars and researchers, regardless of their scientific field of work, are all fast in responding to a crisis, and not just in terms of epidemiological or sanitary aspects. The present pandemic has reopened discussions on ecological revolutions, on discontinuities and ruptures in different historical times, on nature commodification and early globalizations and on early contacts and confrontations between different systems and societies (Brito 2020; Cariño 2020). It has brought new insights and debates on states of dissolution and disruption, of paused life, of paradoxes and contradictions (Vital 2020), and stressed the importance of creating conditions for free access to information on these issues (e.g., *Pandemics in Context*).[3]

Today, thousands of experts and activists from many organizations and different countries are urging for a new 'post-pandemic normality', calling for 'regionality' and empathy, for reducing working hours by calling into question our 21st century understandings of 'being productive' as well as the need for putting life and solidarity at the center of the economy.[4] It is argued that this crisis has highlighted the structural weaknesses of a capitalist economy obsessed with growth, a health system paralyzed by years of austerity, a philosophy that believed in green growth supported by the illusory possibility of unlimited development without degradation. Can we really have a sustainable way of living within the capitalist model? This is one of the questions of the moment. Whatever the answer might be, now is the time to reflect on transition processes into a postcapitalist model while facing the never-ending ill effects of humanity's impact on the planet (Santos 2020).

Over the past few years, greater attention has been focused on these issues, though it is difficult to find consensus or a common path between the various actors in the field; sometimes solutions and agreements are not encountered even among those who promote new paradigms and models for coexistence between humans and nature. For instance, conservationists seem not to agree even on future 'utopian' landscapes. The 'new conservationists' accept the existence of human-dominated habitats where nature is seen as a bundle of ecosystem services, whereas traditional conservation tries to restore 'natural historical environments'—in the sense that all living and nonliving things occur naturally—where biodiversity is protected for its own sake (Keulartz and Bovenkerk 2016). In order to unify these different approaches, a new paradigm of the concept of nature should perhaps evolve. As advocated by Marris (2011), we must correct our habit of "thinking that nature is something 'out there' and far away", when in fact it is everywhere. "Pristine nature is now a dream", and it cannot continue to "blind us" although changing our ideas is not easy (Marris 2011, p. 3). In the same way, concepts of sustainability, of wild and domesticated animals, of agency attributed to animals or to non-humans—see the recent studies on

---

[3]　Pandemics in Context. Multimedia Library Collection. Environment & Society Portal, http://www.environmentandsociety.org/mml/pandemics-context (accessed on 1 May 2021).

[4]　See recent posts by the group NiCHE—Network in Canadian History & Environment (niche-canada.org) (accessed on 1 May 2021).

intelligence and behavior of plants (Mancuso 2018)—starting, at least, from Latour (1993, 2009), are still the subject of heated debate at scientific, social and philosophical levels.

The environmental humanities are one of the most powerful means that we can use to integrate these various concepts of landscape and, going beyond academia's divisions, reach other targets in the civil, political and economic world. For instance, Iovino et al. (2018), speaking of Italian landscapes that have been progressively destroyed or irremediably modified due to 'environmental crimes', shows a concrete example of how an interdisciplinary reinterpretation can be achieved and how this approach can easily reach different publics and targets. Thus the decades-long culture of Italian environmental resistance, which has its roots mainly in the natural sciences, and also the more recent post-human movements, can now see in the environmental humanities a new ally that, as Iovino et al. (2018) claim, gives them another voice.

Being located within a historical perspective, as environmental humanities are, allows both for an understanding of past solutions to similar problems and for an awareness that worries about the degradation of nature were not the same as those that we as individuals, and humankind in general, are experiencing today. In fact the speed of change and the impact of new technology have both changed dramatically between, for instance, the Early Modern Age and the present day. We need narratives that show us what has happened, that permit us to understand the issues we are facing, making room for future paths to be followed. What we need, as Christof Mauch (2019) puts it, "is a deep understanding of our relationship with the environment, alongside critical thinking, and hopeful visions" or again, in the words of Santos (2020), to create conditions for a human redefinition and "re-existence". Moreover, it is indispensable today to address the interaction between humans and the non-human, the more-than-human, the other-than-human world (Haraway 2015; Henry 2018), bringing to the discussion a broad understanding of landscape [5].

The need to rethink our relationship with the rest of the planet obliges us to develop new and efficient solutions to the serious problems that we are facing today. The discussion around the multiple concepts of landscape, interpreted as the result of complex and multifaceted interconnections between us and the natural world, could open up a fruitful discussion on current environmental topics. As also stressed by other authors (e.g., Farina 2021) people and natural process should be finally "considered as inseparable ingredients of the earth's complexity". Environmental humanities and interdisciplinary and postdisciplinary approaches to landscape are of enormous help in our effort to rise above divisions in academia in order to focus on urgent topics and reach different targets in the civil, political and economic world.

**Author Contributions:** Conceptualization, investigation, writing—original draft preparation, writing—review and editing: A.C.R., C.V. and C.B. All authors have read and agreed to the published version of the manuscript.

**Conflicts of Interest:** The authors declare no conflict of interest.

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
