# Peer review of "Shaping Landscapes: Thinking On the Interactions between People and Nature in Inter- and Postdisciplinary Narratives"

_humanities, doi:10.3390/h10020075_

Round 1

Reviewer 1 Report

Very interesting article with a vast and articulated bibliographic review.

Interesting and actual reflection.

Author Response

Many thanks for your comments and the positive evaluation of our manuscript.

Reviewer 2 Report

The essay intends to link the articulated and multidisciplinary reflection on the concept of landscape with the most current frontiers of the relationship between man and world. The text is based on an extensive and updated bibliography (perhaps the profound reflections on culture and landscape of cultural geography could be more closely related). The text is clear and well argued, the problems and reflections are proposed in an orderly manner. The image is consistent with the reasoning. The essay can be published.

Author Response

Thank you very much for you comments and the positive evaluation of our manuscript.

Reviewer 3 Report

Shaping Landscapes" is a well-written summary of some recent scholarship on landscapes, mainly theoretical and methodological studies with a strong Eurocentric bias. Although works on Africa, Latin America, and elsewhere are cited, the prism through which the author(s) write is distinctly Western European, and thus of limited interest to readers elsewhere. There is a sizable body of writing, theoretical as well as empirical, on North American and East Asian landscapes that is barely touched on in this manuscript. The Berque and Kim references barely scratch the surface. The manuscript, which is basically a literature review, aptly illustrates the principle identified by Michael Oakeshott years ago that practical action comes first, then theory follows after. Without grounding in the empirical elements that comprise landscapes, the theoretical/methodological principles advocated in this manuscript seem detached from human and nonhuman experience, limiting their value to an understanding of the world we live in. There is no harm in publishing the manuscript, but its impact is limited by its geographical and ideological limitations.

Author Response

Many thanks for your comments, we agree with you that the manuscript is perhaps more European oriented, but it was not our intention to present a comprehensive review of the landscapes concepts and how they were applied worldwide. Similarly we are aware that the rapid expansion of environmental humanities cannot be embraced by a short article. Our work is not a review paper, and our idea was to show our experience and open a new debate on this topic by bringing together the humanities, and the multiple visions of landscape to unite once separate visions and disciplines.

We have outlined that aspect now in the end of the introduction and added a couple more references.

The initial idea was also to present our Special Issue on Landscape to justify the presence of very different works that however converge in a single point, understanding how human and nonhuman interact, in the past and in the present.